# Green Recycling for Polypropylene Components by Material Extrusion

**DOI:** 10.3390/polym16243502

**Published:** 2024-12-16

**Authors:** Roberto Spina, Nicola Gurrado

**Affiliations:** 1Dipartimento di Meccanica, Matematica e Management, Politecnico di Bari, 70100 Bari, Italy; nicola.gurrado@poliba.it; 2Istituto Nazionale di Fisica Nucleare (INFN)—Sezione di Bari, 70100 Bari, Italy; 3Consiglio Nazionale delle Ricerche—Istituto di Fotonica e Nanotecnologie (CNR-IFN), 70100 Bari, Italy

**Keywords:** material extrusion, recycling, material testing, polypropylene

## Abstract

High volumetric shrinkage and rheological behavior of polypropylene (PP) are the main problems that make material extrusion (MEX) uncommon for this material. The complexity is raised when recycled materials are used. This research covered different aspects of the MEX process of virgin and recycled PP, from the analysis of rough materials to the mechanical evaluation of the final products. Two types of virgin PP (one in pellet and the other in filament form) and one recycled PP were analyzed. Thermal characterization and rheological analysis of these materials were initially employed to understand the peculiar properties of all investigated PP and set filament extrusion. The 3D parts were then printed using processed filaments to check fabrication quality through visual analysis and mechanical tests. A well-structured approach was proposed to encompass the limitations of PP 3D printing by accurately evaluating the influence of the material properties on the final part performance. The results revealed that the dimensional and mechanical performances of the recycled PP were comparable with the virgin filament commonly employed in MEX, making it particularly suitable for this application.

## 1. Introduction

The automotive market constantly expands, with new increments due to car electrification. The sales growth in electric vehicles (EVs) in the global contracting car markets dipped by 3% from 2021 to 2023. Electric car sales exceeded 14 million last year, bringing their total number on the roads to 40 million. The total fleet of EVs will grow from 40 million in 2023 to 240 million in 2030, with an average annual growth rate of 35%. EVs will account for over 18% of the road vehicle fleet by 2030 [1]. Plastic recycling should be improved to compensate for this expansion in the passenger car market. Many factors, including the generation of more waste, enlarged extraction of petrol-based materials, excessive energy consumption, and European Union requirements, force the implementation of the circular economy in the car market. Moreover, a strong positive correlation is recorded between the level of waste recycling in a country and its rate of product and process innovation [2].

Automotive polymers are well known for their high performance and versatility, satisfying the rigorous demands of this sector’s standards and regulations, the innovation demand, and the fuel consumption reduction. Polypropylene (PP) and polyamide (PA) are the most semicrystalline plastics used in car interiors, exteriors, and under hoods, thanks to the possibility of realizing highly complex parts. They are characterized by great chemicals and heat resistance, an easy way to realize moldable parts. The lifecycle of automotive materials starts with raw materials, turned into various products and components. At the end of life, car plastic components are discarded as waste. Since PP and PA are primarily used for multiple products, they are widely discarded in ecosystems [3]. Recycling and upcycling them is an effective solution for reducing waste. Recycling reduces the use of raw materials and wastes through a closed-loop system, whereas upcycling adds value to plastic waste, obtaining a higher-value product. The difficulty and cost of classification and separation according to material properties hinder recycling and upcycling [4]. Identifying the chemical and physical modifications occurring after recycling is needed to assure part quality and avoid risks of deterioration and failure. The presence of contaminants negatively influences the material properties, mainly when multiple waste sources are employed. Due to the strict material quality and performance requirements, this aspect limits recycling and upcycling potentials in the automotive sectors [5].

Material extrusion (MEX) is an Additive Manufacturing (AM) process in which a polymer is selectively dispensed through a heated nozzle and deposited melted material layer by layer to build a 3D object [6]. The process is characterized by good speed, low precision, and the possibility of using several raw thermoplastics [7]. In the case of a semi-crystalline polymer such as pure PP, high volumetric shrinkage and warp deformation occur due to its crystallization degree [8]. The morphology of a semi-crystalline polymer is a direct effect of the process parameters (e.g., deposition speed), product design (zones or areas with high thermal gradients), and the printed part shape and dimensions [9]. A good printed part quality can be achieved by also solving adhesion with the printing bed. The reduction of high crystallinity degree and volumetric shrinkage of PP can be obtained by blending or copolymerizing with other polymers [10]. A high-quality filament characterized by a smooth surface, a precise circular cross-section, and a constant diameter along all sections reduces issues related to inhomogeneous deposition, blocks of the pulling system, and nozzle clogging [11].

Despite the advantages, MEX generates large amounts of waste from failed components or rejected support structures. Filaments are often extruded using granulates transformed into a homogeneous material with defined parameters, such as temperature and rotation speed [12]. MEX thermoplastic waste recycling shows a decrease in viscosity, molecular weight, and thermal and mechanical properties after one or multiple rounds of recycling, which may be compensated by adjusting the values of process parameters [13]. The degradation also affects the blends’ compatibility [14]. Recycling for the same use reveals essential safety and technological issues in processing waste materials similar to their virgin counterparts [15]. Different combinations of process parameters in MEX lead to various levels of fabrication quality and mechanical properties. MEX induces the rearrangement of polymer molecules and changes in the thermo-mechanical material properties [16]. Process parameters such as infill density, nozzle temperature, raster angle, and layer thickness affect mechanical properties like tensile strength, yield strength, modulus of elasticity, and elongation of MEX parts [17]. Two important factors should be considered in reducing the mechanical properties of recycled PP (rPP). The first is that rPP is exposed to demanding conditions for extended periods, causing property decay. The second is that rPP is subjected to thermo-mechanical degradation for high shear forces and temperatures in an oxygen-deficient atmosphere [18]. Because rPP is susceptible to property deterioration, its application in the core layer of sandwich-structured multi-layer products is a promising method without limiting component aesthetics and other surface-related performance features [19]. Additional solutions to improve properties are adjusting the MEX process parameters and developing materials with suitable thermo-rheological properties. The printing conditions are optimized for rPP or complemented with other processes in the first approach. In the second one, rPP favors interfacial fusion by blending a high-viscosity polymer with a lower melt viscosity [20].

In this article, mechanical testing samples were fabricated with MEX from virgin and recycled polypropylene in pellets and filaments. This study followed the process chain in each step (preparation, characterization, and fabrication) and evaluated the performance of MEX recycled parts compared with virgin parts. One primary objective was to characterize the printability of recycled polypropylene by determining the presence of contaminants and their effect on thermal properties, accepting the addition of materials different from PP for a percentage less than 10%. This research improved knowledge about how changes in material properties impact the printability, visual, and mechanical properties of MEX parts, highlighting the potential of recycled propylene in circularity for sustainability. The quality was also appraised using an optical sensor for filament measurements and visual inspection of MEX reference parts. Tensile specimens printed from these filaments were used to evaluate the mechanical properties of the different materials, expecting a performance variation in modulus and strength of 1–5%. This study gives insight into the potential for practical MEX filament and printing with feedstock gathered from waste polypropylene, pointing out the good performance of recycled polypropylene compared to virgin materials.

## 2. Materials and Methods

Figure 1 reports the framework used to analyze the thermo-mechanical properties of the investigated PP/rPP, starting from the raw materials and ending with comparative results. The arrows in the figure represent the sequence and knowledge improvement after each step; details are fully described below.

The raw PP materials were the Ultrafuse^®^ PP (BASF 3D Printing Solutions GmbH, Heidelberg, Germany), the PP 505P (SABIC Europe, Sittard, The Netherlands), and the recycled PP (Recuperi Pugliesi srl, Modugno, Italy). The materials were supplied in 1.75 mm diameter filaments (BASF) and pellets (SABIC and Recuperi Pugliesi). The recycled PP (rPP) had properties similar to those of the PP 505P.

Differential Scanning Calorimetry, Rotational Rheometry, and Fourier-transform infrared (FT-IR) spectroscopy were used to characterize the above materials. The thermal analysis used a silver furnace with a high-temperature DSC 404 F1 Pegasus (Netzsch-Gerätebau GmbH, Selb, Germany). Samples weighing 10 mg were put in pure aluminum pans under a nitrogen atmosphere from ambient to the maximum temperature of 250 °C. The heating and cooling rates were set to 10 °C/min. Rheology was determined using a rotational rheometer HAAKE Mars III (Thermo Fisher Scientific Inc., Waltham, MA, USA) in parallel-plate geometry. For stable torque conditions, 20 mm diameter platens were employed in the 200–250 °C temperature range under a controlled shear rate of 0.1–50 s^−1^. A 0.5 mm gap was set during tests in a nitrogen atmosphere, repeating measurements five times. Possible thermal degradation during the tests was evaluated before and after the tests. A Fourier-transform infrared spectroscopy (FT-IR) device model 4200 (Jasco Co., Tokyo, Japan) was used for the material characterization in Attenuated Total Reflectance (ATR) mode with 64 scans at a resolution of 4 cm^−1^. Several FT-IR-ATR measurements were carried out at arbitrary points for each sample, and an average spectrum was computed. The spectroscopic transformation to absorbance, baseline correction, and standard normal variate (SNV) conversion were performed on the spectra.

The fabrication of filaments and following use for realizing specimens were carried out with a Next 1.0 filament-making machine (3devo B.V., Utrecht, The Netherlands) and a 3D printer Bambu Lab X1 Carbon (Shenzhen Tuozhu Technology Co., Shenzhen, China). A 1.75 mm filament was directly produced from pellets. The Next machine was a single-screw extruder with four independent heating zones, an integrated filament positioner, and a spool winder. The temperatures of each heating zone, the screw speed, and the fan speed were accurately controlled. After the nozzle exit, a customized air fan system was designed and installed to homogeneously cool the extruded filament, avoiding the section shrinkage for a high crystallinity degree. A system with two counter-rotating wheels guided the extruded filament, guaranteeing the desired diameter by adjusting the proper tension with speed control. The X1 Carbon printed the specimens designed with Fusion 360 (Autodesk Inc., San Francisco, CA, USA). Spools of different PP and rPP filaments were placed in the Automatic Material System (AMS) at room temperature and controlled humidity. A 0.4 mm nozzle was used to balance speed and quality. Standard printing parameters were a layer height of 0.2 mm (balance between print quality and speed), three top layers, three bottom layers, three walls, and a raster orientation of ±45°. The adhesion of the first layer was prepared with a brim pattern and using a Magigoo PP glue, specifically for this use, for material adhesion to the Polyetherimide (PEI) plate. Several part geometries were realized and investigated. Initially, simple edge calibration test cubes (edge length equal to 20 mm) were printed with the X, Y, and Z letters written on the orthogonal faces to measure the main dimensions against the expected ones. Moreover, tensile test specimens (dog-bone shape) were fabricated with geometry based on ISO 527-2:2012 type 5A [21]. A variable infill was adopted, such as 15% with a cubic pattern for dimensional analysis cubes and 100% for tensile specimens.

The following phase (inspection) assessed the specimen’s precision and mechanical strength. Cubes of the different materials were measured using a non-contact technique by using a 3D optical scanner, GOM Atos Q 8M (Carl Zeiss GOM Metrology GmbH, Baden-Württemberg, Germany), consisting of a structured narrow-band blue light projector and two cameras capturing fringes based on the stereo camera principle. The scanner was equipped with 100 MV lenses, having a measuring volume of 100 × 70 × 60 mm^3^ and certified accuracy (5 μm of Length Measurement Error assessed according to the VDI/VDE 2634—Part 3). The scanner measuring distance of the MV 100 version was 100 × 70 mm^2^, with 8 million points acquired during each single scanning. Comparing the whole point clouds with the nominal geometry of the specimen allowed the macro geometry and deviations to be evaluated. Mechanical tests on dog-bone specimens were conducted on an ETM type A universal single-column testing machine (Wance Testing Machine Co., Ltd., Shenzhen, China) with a 5 kN load cell. All measurements were conducted with a 5 mm/min transverse speed in a temperature-controlled environment, with a maximum variation of ±0.5 °C from the ambient temperature.

## 3. Results and Discussions

### 3.1. Material Characterization

Table 1 reports the resulting thermal properties, and Figure 2 shows the thermographs of the investigated materials. The results were in agreement with the literature [22].

The melting peak *T_melt_* was 170.2 °C and 131.8 °C for the SABIC and BASF, respectively. The significant difference in these values was associated with the specific polymer nature. The SABIC was a homopolymer, while the BASF was a copolymer. Copolymer PP was mainly used for its low melting point and high fluidity, thanks to catalysts changing the base polymer’s properties [23]. The BASF filament needed lower melt energy than SABIC. The thermograms of the rPP revealed a contaminant that introduced an additional melting peak at lower temperatures. The colorant pigments (black and white) were transparent to the thermal analysis. This contaminant was a low-density polyethylene (LDPE) because of its melting peak *T_melt_* of 128.6 °C and enthalpy value. The LDPE content was computed as a minor of 10%, using a well-known procedure [24]. To calculate the degree of crystallization *X_c_*, the melting enthalpy *ΔH_melt_* was divided by the melting enthalpy of 100% crystalline material, equal to 209 J/g [21]. More energy was required to melt the rPP filament than the pure PP filament due to an increase in the melting peak and enthalpy caused by the rise in crystallinity [25].

The FT-IR analyses confirmed the information gained from the DSC. The SABIC PP and LDPE spectra (Figure 3) were initially evaluated, identifying the functional group absorption bands in Table 2. PP was made from the propylene monomer, where the C=C bond reacted to form chains, whereas the LDPE was made from the ethylene monomer, where the C=C bond split open to form chains. PP was characterized by CH_3_ stretching peaks at 2951 and 2870 cm^−1^, CH_2_ stretching at 2915 and 2839 cm^−1^, and CH_3_ bending at 1458 and 1377 cm^−1^. LDPE was identified by the CH_2_ stretching peaks at 2920 and 2852 cm^−1^, a C-H bending peak at 1458 cm^−1^, and a CH_2_ rocking peak at 716 cm^−1^. The two polymers had very similar chemical structures, but this behavior did not guarantee good interfacial adhesion or polymer blend. These two polymers were thermodynamically immiscible due to the positive Gibbs free energy of the system, causing the creation of a separation interface and weakening the interfacial adhesion between them. The extra energy at the interface separated them, leading to interfacial failure and lowering the mechanical properties. This was the case with polyethylene (PE) and polypropylene (PP), which were incompatible, although they had similar structures [26]. The two rPP spectra were equal and perfectly overlapped, with a negligible influence of the colorant pigment on the chemical structures. Figure 4 shows the virgin PP and rPP spectra, typical of a polyolefin, with some specific differences. A broad absorption band with multiple peaks was detected in the 2800–3000 cm^−1^ and 1400–1500 cm^−1^ ranges. Some peculiarities were observed in the rPP spectrum. The relative intensities of bands attributed to CH_2_ groups at 2920 cm^−1^ and 2839 cm^−1^ compared to CH_3_ groups were relatively high, revealing the polyolefin cross-contamination with LDPE. The peak at 1715 cm^−1^, associated with the carbonyl band, was an indicator of thermo-oxidative degradation during PP recycling. A pronounced peak at 716 cm^−1^ also indicated LDPE contamination, as expected in the recycling process.

The polymer viscosity was significant in the MEX process. The rheology was mainly influenced by molecular weight distribution and polydispersity. The material behavior was assessed by measurement. The rheological properties related to the flow behavior were analyzed using a rotational rheometer. The storage *G*′ and the loss *G*″ moduli were computed as a function of frequency ranging between 10^−1^ to 50 rad/s, with 1% strain, 0.5 mm gap, and under a constant temperature during each experiment. The time–temperature superposition (TTS) principle was employed to compute the master curves for each specific temperature covering a broader frequency range. Master curves of *G*′ and *G*″ were particularly helpful in understanding the rheological behavior. The superposition process separated the two main variables—time and temperature—upon which the viscoelastic properties depended, expressing the properties as a single function for each. Based on the *G*′ and *G*″ master curves, information on the sample’s complex viscosity *η^*^* was computed. Figure 5 shows the master curve of *η^*^* at 250 °C of the three polymers as a function of shear rate ω, estimated to simulate the printing conditions. At 250 °C, the polymers still had several entanglement points, and extensive degradation due to mechanical chain breaking did not occur [27]. The data consisted of measured rheological shifted data (scatters) with TTS and curves fitted using the Carreau–Yasuda model. The viscosity decreased with increasing shear rate, expressing a typical pseudo-plastic fluid behavior (shear thinning) essential for the filament’s good processability through the nozzle. The polymers exhibited a shear thinning behavior (decreasing viscosity/decreasing the slope of the shear stress curve with increasing shear rate), consistent with their thermoplastic nature. Polymeric chains were arranged in a somewhat random or entangled configuration at low shear rates, resulting in high viscosity. As the shear rate increased, the polymeric chains experienced a higher level of deformation or strain, leading to disentanglement and alignment of the polymeric chains in the flow direction while reducing the viscosity. The SABIC was more viscous than the other two filaments because the methyl side group in the homopolymers raised the disentanglement to the polymer chain and caused a reduction in the chain mobility [28]. The presence of PE in rPP had a beneficial effect on the filament rheological behavior at low shear rates because its viscosity resulted in lower viscosity than SABIC and BASF, thanks to the better flowability of PE. At higher shear rates, the effect of PE was less evident, with the appearance of a yield stress behavior in the low-frequency range. This behavior, a consequence of an optimal distribution and dispersion of the PE in the PP, was beneficial for 3D printing by giving the material the necessary stable conditions during deposition, also allowing its shape preservation.

### 3.2. Filament Making

The standard filament had a diameter of 1.75- or 2.85 mm. Different values in excess or defect could cause obstruction or intermittent extrusion [29]. The extrusion parameters, environmental conditions, and the material state at the processing time influenced the filament-making process. Regarding environmental conditions, the tests were performed at a temperature of 24 °C and relative humidity of 34%, operating in strict ranges with a maximum variation of ± 2%. Filaments, 1.75 mm in size, were produced using the same parameters for all pelleted materials. An electric motor was directly connected to the extrusion screw. The material entering the hopper (Figure 6) was then heated from four heaters around the barrel (H4, H3, H2, and H1), each monitored with a K-thermocouple. Polymer melting was realized in conjunction with the friction-generated heat. An adaptive puller system adjusted the speed for consistent diameter control while maintaining a constant tensile force synchronized with the spooler rotation velocity. The optical sensor between the nozzle and puller measured filament diameter to a max precision of 43 μm. To achieve effective filament cooling after nozzle exit, starting from suggestions of the machine supplier, 3D-printed conveyors were applied to the side fans to distribute the airflow around the hot sections better, avoiding ovalization. The parameters adopted for the filament extrusion were the outcomes of a trial-and-error approach using a multistep procedure. The temperatures of the four heaters were raised to set point temperatures, and a small quantity of pellets was inserted into the hopper, checking that the extrudant came out regularly from the nozzle. After reaching stable conditions, further pellets were added to the feeder, and the filament-making process started, avoiding too tenacious material, which broke immediately once put under tension. The initial values based on a report about the recycling of PP given by 3devo support [30] were initially used, but the diameter variations were high. The grey points in Figure 6 are the diameter values of SABIC PP and rPP, measured from the machine sensor with an acquisition frequency of 1 Hz.

New stable conditions were reached by increasing the temperature of all heaters and fan speed, based on experience cited in the literature [11]. These increases generated smoother surfaces and improved the filament quality. The final values were identified after appraising the most minor variations in diameter dimensions (Table 3). The blue lines in Figure 7 are the SABIC and rPP diameter values compared with the BASF filament diameter measured with the filament-making sensor.

The variations of the rPP diameter were higher than the SABIC and BASF, with a median value of 1.752 mm. These values were sufficient to avoid filament clogging and guarantee stable printing conditions. The reason for these results was associated with the difference in material viscosities. Materials with lower viscosity at processing shear rates, whether derived from a lower average molecular weight or broader molecular weight distribution, flowed more quickly, requiring less pressure and energy to extrude. A high-viscosity polymer at low shear rates was advantageous in these cases. A broad molecular weight distribution was beneficial when the viscosity was low in the high-shear portions of the process. A high-molecular-weight polymer also exhibited excellent toughness and chemical resistance. From the rheological characterizations, printing parameters should be selected to exploit the polymer melt’s shear-thinning nature by enabling extrusion at lower forces due to reduced viscosity and limiting the orientation-induced effects affecting the crystallization process by controlling the print speed.

### 3.3. Three-Dimensional Printing

The quality of a 3D print was not only defined by the properties of the filament used but also by the processing parameters such as nozzle and build plate temperatures, layer height, raster orientation, and printing speed. Similarly to the filament-making process, different tests were performed to identify the 3D printing parameters suitable for all filaments, varying nozzle and build plate temperatures, and then printing speed. The precise control of temperatures throughout the process was crucial because temperatures were the most critical parameters for successful 3D printing [31]. Initial trials ranged the nozzle temperature between 200 and 250 °C, similar to rheological studies, and the build plate between 50 and 100 °C. At the end of the trials, the best nozzle temperature was 240 °C, limiting excessive warping for higher temperatures while filament clogging for lower temperatures. Concerning the built plate, the temperature was set to 80 °C because lower values led to poor platform adhesion, whereas higher temperatures caused layer delamination and excessive warping. Lower values of bed temperature led to warping of the part, as the difference in temperature between the extruded material and the printing bed must be kept as low as possible to reduce the shrinkage related to the cooling of the material. Additional parameters were a maximum flow rate of 6 mm^3^/s for a printing speed of 60 mm/s. The semi-crystalline nature of PP caused shrinkage of the printed parts, leading to distortion and decoupling from the print bed. The printing speed of the first layer was lowered to 40 mm/s to increase the adhesion and ensure the first layer’s accuracy. To further minimize warping phenomena, the closed printing chamber was pre-heated to 60 °C before realizing the parts to reduce the thermal gap between extrusion and the surrounding environment. In addition, the poor adhesion of polypropylene to the print surfaces required Magigoo PP glue. The most significant difficulties encountered in the printing phase occurred with lab-produced filaments, especially with SABIC PP and rPP, due to the slight instability of the extrusion process and, therefore, the non-constancy of the filament diameter in a few sections. At some points, the material section was too large to pass easily through the extruder throat. Failures during printing occurred even with purchased filament (with a more constant diameter), as PP tends to jam (obstruct) due to print retraction. No other parameters, including layer height and raster orientation, were further modified to achieve homogeneous sets among filaments. Figure 8 shows the geometrical model dimensions with the slicer software(version: 1.10.0), and Figure 9 shows the produced specimens. The effect of build chamber temperature on crystal growth and mechanical property enhancement needs further investigation.

### 3.4. Inspection

The inspection consisted of capturing, filtering, and aligning the point cloud of each specimen and then evaluating the geometrical deviations from its CAD model. A non-invasive capture process was conducted with the GOM scanner sensor to obtain 3D point sub-clouds of cube specimens with the support of an incremental rotational encoder. Three-dimensional point sub-clouds were expected to contain large amounts of data with high complexity since the more detailed and accurate the representation of the real object was, the higher the total number of sub-cloud points. A noise reduction was performed on these sub-clouds to remove erroneously generated points introduced by the capture operation, followed by a filtering operation to reduce each sub-cloud’s total point number without losing accuracy. The filtered point sub-clouds were aligned using the ZEISS Inspect^®^ software (Zies Inspoecvt 2023) to achieve the whole cloud of the specimen with the Local Best Fit algorithm after a fine-tuned registration procedure. The surface deviation between the final point cloud and the specimen CAD model was measured during the evaluation stage. The colors gave a quick overview of areas below or above tolerance with the CAD model, ranging from the blue color when below tolerance to the red color when above tolerance. Figure 10 shows the evaluation deviation maps in the ISO view for the MEX specimens realized with the different filaments. No influence of color pigment in the rPP filaments was detected on the inspection results. At first glance, the cube made with BASF was the most accurate, followed by the rPP and then SABIC. The level of detail was suitable for all specimens, and the deviation was in the range of ±0.5 mm for a cube edge length equal to 20 mm.

However, the plane and side views (Figure 10) evidenced a particular material’s effect on the deposition plane. Residual stress induced by the MEX process led to distortion and possible de-layering problems, seriously affecting the cube shape and final dimensions. Warping mainly occurred in the first layers adhering to the printing bed. The upward curvature of the individual layers could be so high during printing that the model could detach from the plane, causing the printing process to fail. Focusing on the filaments, the rPP had the lowest deviation from the plane, limited to the cube corners. In contrast, the SABIC had the most significant variations, sometimes with large zones of detachment from the machine platen. These variations depended on previously investigated material properties, such as melting/cooling peaks and viscosity. The SABIC was more viscous and had higher crystallinity than the other two filaments. The high crystallinity caused a rapid shrinkage during cooling because of the transition from a disorganized semi-solid state to an organized crystalline solid state. The BASF and rPP had similar viscosity, even with different crystallinity, making the last material attractive for fabricating MEX samples.

### 3.5. Mechanical Testing

Tensile tests evaluated the specimens’ strength of the filaments. Mechanical tests on dog-bone specimens were conducted with a 5 mm/min transverse speed in a controlled environment, with a maximum temperature variation of ±0.5 °C from the ambient and humidity between 20 and 35% RH. The initial cross-section of each specimen was measured before testing using a digital micrometer. The maximum strain set for each test was 5%. Three repetitions for each material were carried out, and the average results were reported in the stress–strain graph (Figure 11). All specimens broke ductile, considering the plateau after the yield point with a significant stress drop [32]. The yield point was identified as the first point on the stress–strain curve at which an increase in strain occurred without an increase in stress.

rPP and BASF were characterized by high-elongation and tensile-toughness values (area under stress–strain curve). On the contrary, the SABIC material presented a short elongation with a higher yield value. Table 4 lists the tensile properties of the printed specimens, such as Young modulus, stress at yield, strain at yield, stress at plateau, and strain at break. A portion of the stress–strain curves between 5 and 20 MPa was used for the modulus calculations. The results of the Young modulus and yield stress confirmed the high performance of the SABIC filament due to the high crystallization degree. rPP could be successfully used for MEX due to its similar performance to the BASF.

## 4. Conclusions

Virgin and recycled polypropylene (PP) were used in material extrusion (MEX) polymer processing. The process chain implemented in this research passed through the preparation, characterization, and fabrication of materials, and the performance of MEX recycled parts was evaluated compared with virgin parts.

The thermal and infrared analyses revealed that polyethylene (PE) was a contaminant in the recycled PP (rPP). Due to its presence, increases in the melting peak and enthalpy caused by an increase in crystallinity were recorded, requiring more energy to melt the rPP filament than the pure PP filament. This condition demanded a better setup in filament making. Regarding the visual and mechanical properties of the MEX parts, the recycled PP resin was superior to the homo-polymer PP (SABIC) and comparable to copolymer PP (BASF). Adding PE to PP not only improved the quality of the printed parts, with less warp deformation and adequate mechanical properties, but it also provided the framework for economically MEX-based manufacturing of recycled PP. In conclusion, the reported extensive and in-depth analysis of a recycled PP led to successful material combinations offering less warp deformation and high stiffness.

Further research should be addressed to evaluate the intra-layer resistance of MEX parts and optimize the PP/PE blend.

## Figures and Tables

**Figure 1 polymers-16-03502-f001:**
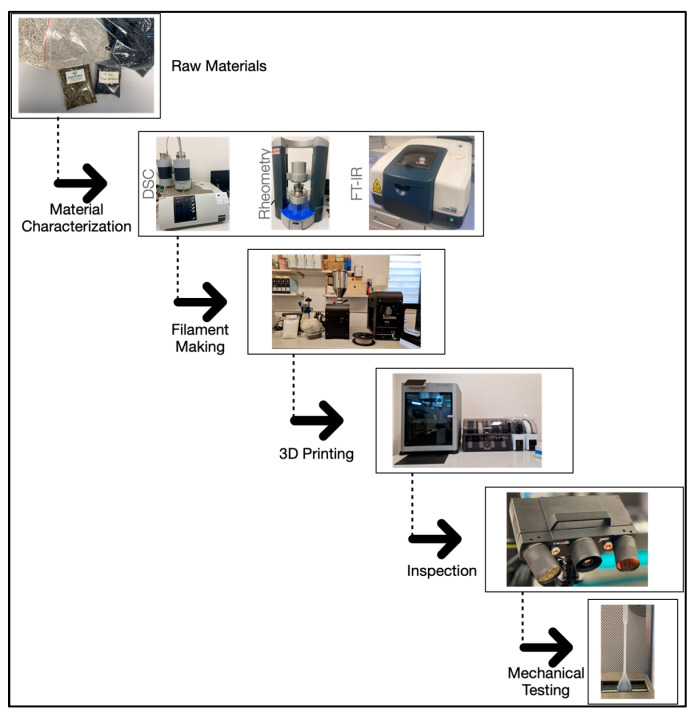
Study flowchart.

**Figure 2 polymers-16-03502-f002:**
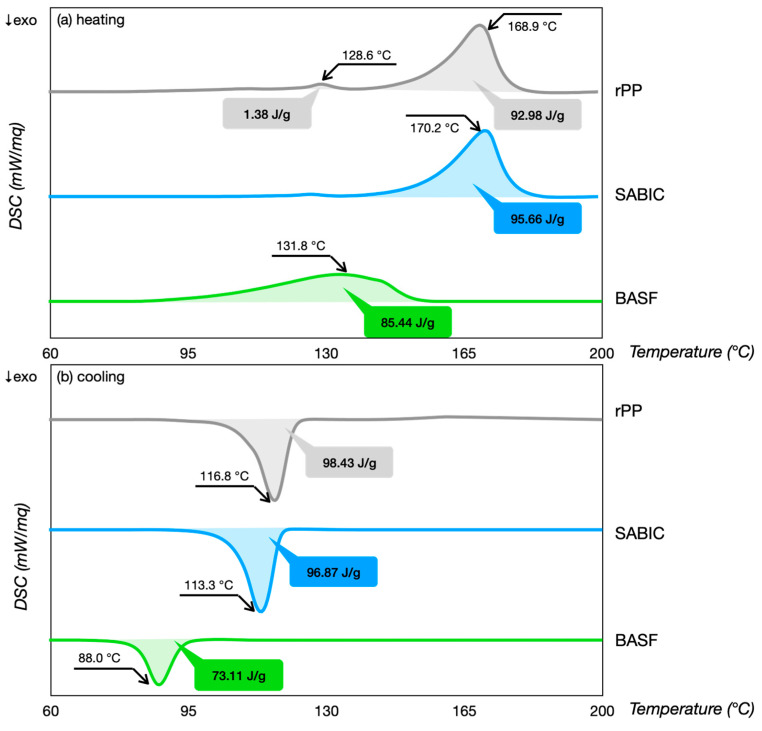
Thermograms of heating (**a**) and cooling (**b**) phase.

**Figure 3 polymers-16-03502-f003:**
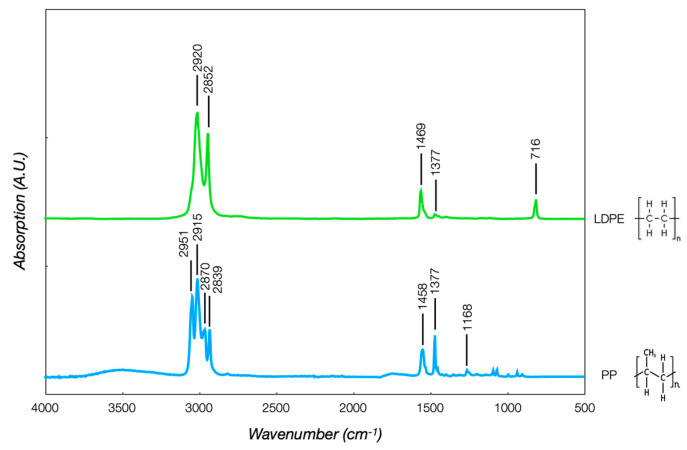
SABIC PP and LDPE spectra.

**Figure 4 polymers-16-03502-f004:**
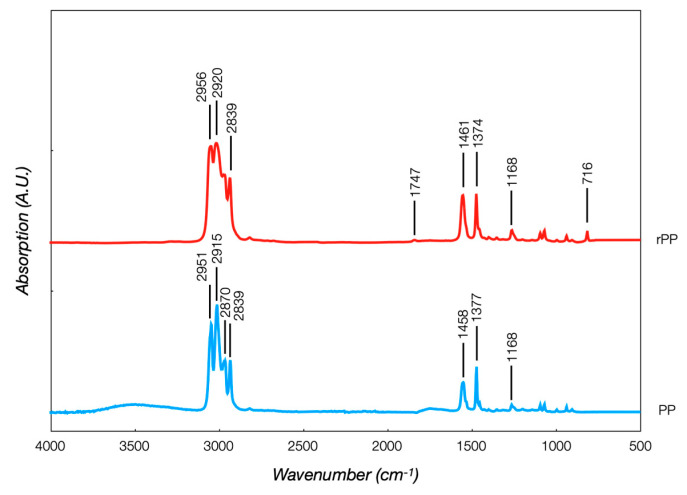
SABIC PP and rPP spectra.

**Figure 5 polymers-16-03502-f005:**
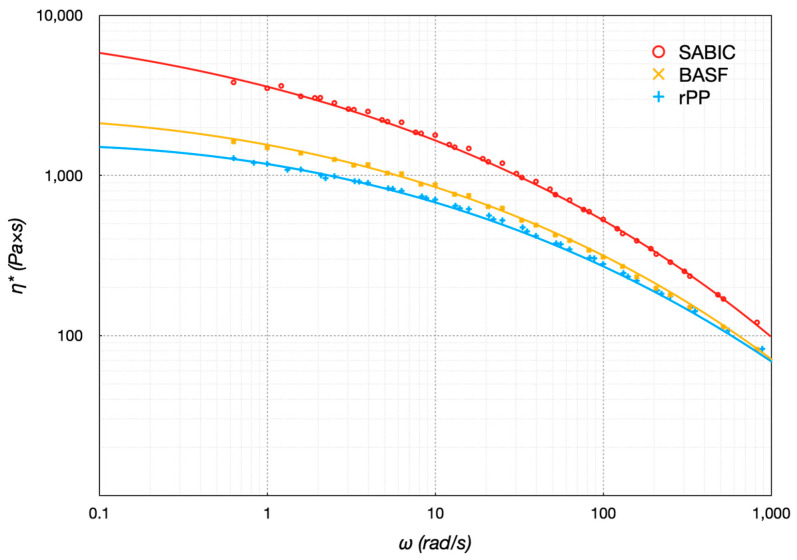
Master curves at 250 °C.

**Figure 6 polymers-16-03502-f006:**
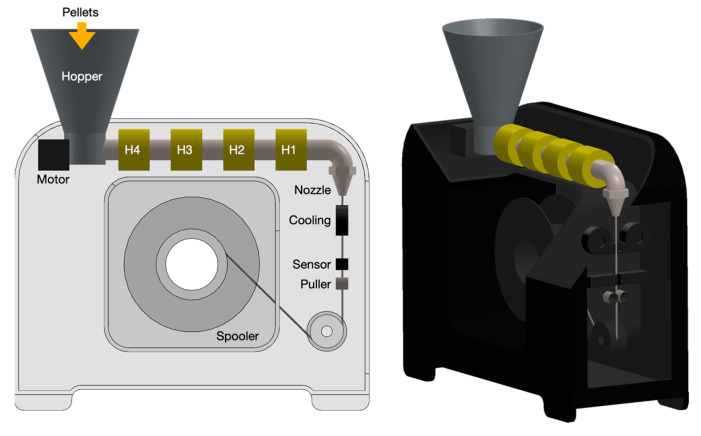
3devo filament maker scheme.

**Figure 7 polymers-16-03502-f007:**
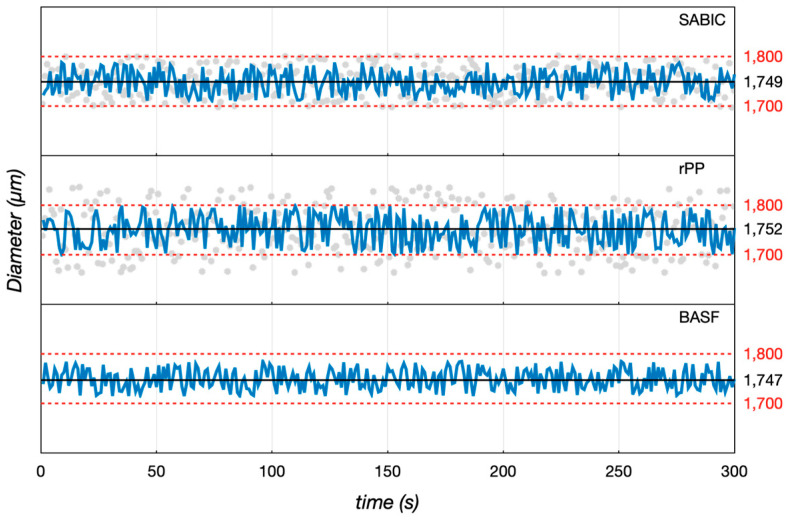
Diameter variations.

**Figure 8 polymers-16-03502-f008:**
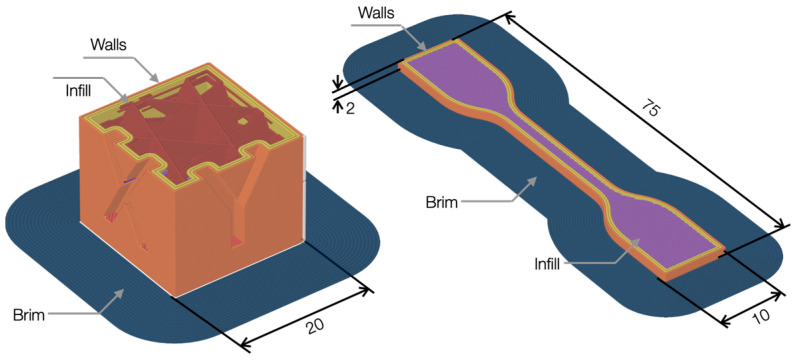
Specimens (all dimensions in mm).

**Figure 9 polymers-16-03502-f009:**
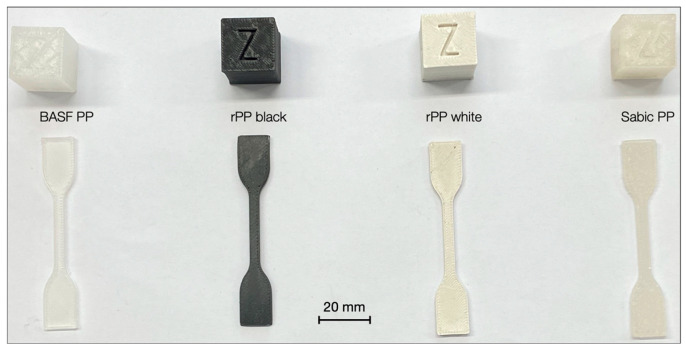
Specimen types.

**Figure 10 polymers-16-03502-f010:**
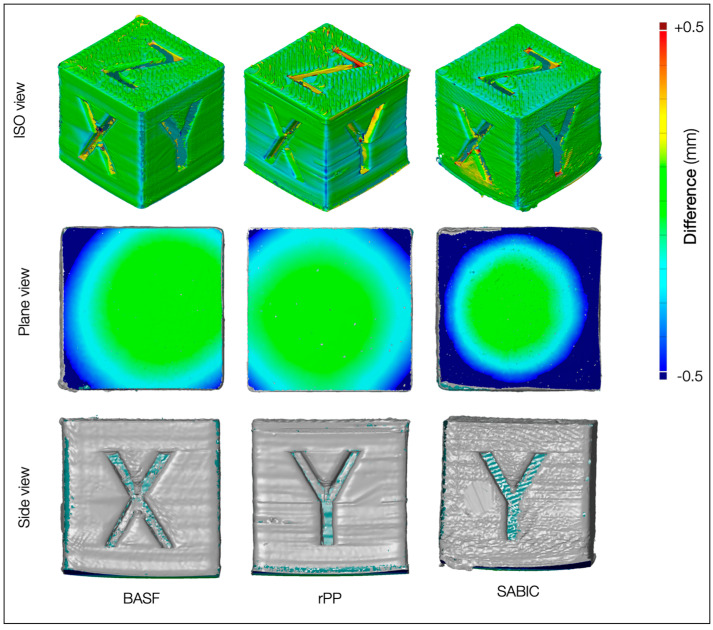
Evaluation deviation map of the cubes (ISO view and plane view) and separation from the substrate (Side View).

**Figure 11 polymers-16-03502-f011:**
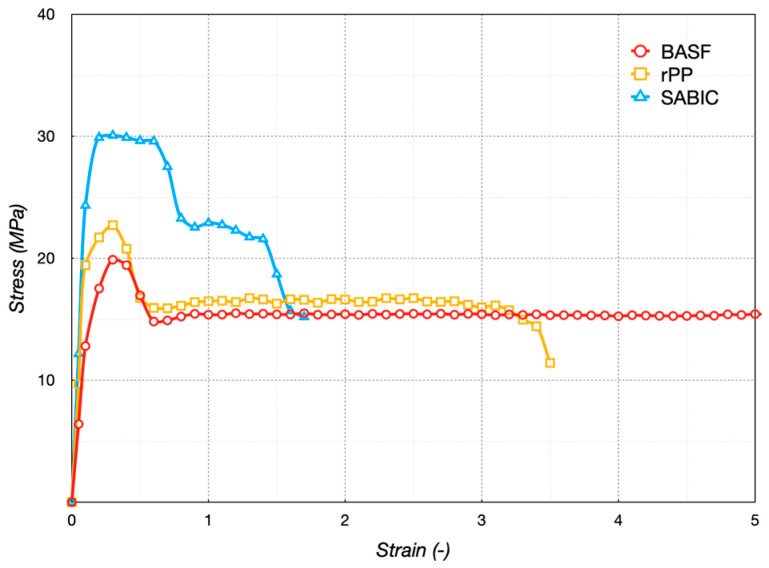
Tensile test results.

**Table 1 polymers-16-03502-t001:** DSC results.

Material	*Primary T_melt_* (°C)	*Secondary T_melt_* (°C)	Δ*H_melt_* (J/g)	*T_cry_* (°C)	Δ*H_cry_* (J/g)	*X_c_* (%)
SABIC	-	170.2	92.98	113.3	96.87	44.48
BASF	-	131.8	95.66	88.8	73.11	45.77
rPP	128.6	168.9	85.44	116.8	98.43	40.88

**Table 2 polymers-16-03502-t002:** Functional groups.

Range (cm^−1^)	Group	Class	Intensity
665–730	C=C bend	alkane	strong
1330–1420	O-H bend	alcohol	medium
1390–1365, 1465–1440	CH_3_ bend	alkane	medium
2840–3000	C-H stretch	alkane	medium

**Table 3 polymers-16-03502-t003:** Parameters of the filament extrusion.

Parameter	H4 (°C)	H3 (°C)	H2 (°C)	H1 (°C)	Screw Speed (RPM)	Fan Speed (%)
Initial values	170	175	175	170	5	20
Final values	190	195	195	190	5	100

**Table 4 polymers-16-03502-t004:** Tensile test results.

Material	Young Modulus (GPa)	Yield Stress (MPa)	Yield Strain (-)	Plateau Stress (MPa)	Maximum Strain (-)
SABIC	1234 ± 49	30.07 ± 2.05	0.32 ± 0.02	22.5 ± 0.4	1.68 ± 0.13
BASF	1180 ± 20	19.88 ± 0.73	0.31 ± 0.02	15.4 ± 0.2	-
rPP	1180 ± 35	22.71 ± 1.22	0.33 ± 0.01	16.4 ± 0.3	3.49 ± 0.07

## Data Availability

Data will be made available on request.

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
