# Peer review of "Green Recycling for Polypropylene Components by Material Extrusion"

_polymers, 2024, doi:10.3390/polym16243502_

Round 1

Reviewer 1 Report

Comments and Suggestions for Authors

Manuscript Title: Green Recycling for Polypropylene Components by Material Extrusion

1. It would be better if the title included more details about the research.

2. It is best to state the most important results of the manuscript quantitatively at the end of the introduction.

3. The research innovation is not highlighted at the end of the introduction.

4. It is better to display Figure 1 hierarchically.

5. The selection criteria for printing parameters such as nozzle diameter and layer height are unknown.

6. It would be better to add the semi-crystalline percentage to Table 1.

7. The selection of parameters in Table 3 requires discussion and reference.

8. It would be better to merge Figures 9 and 10.

9. The values in Table 4 require deviation from the standard.

10. In Figure 9, it can be seen that the samples are not properly attached to the substrate and the separation between the layers is observed. If possible, add a left view and a front view of the samples to Figure 9.

11. It is better to 3D print large-sized samples to better characterize the effects of shrinkage and interlayer separation.

Comments on the Quality of English Language

Check and fix grammatical errors throughout the manuscript. The use of Grammarly is recommended.

Author Response

Dear Reviewer, Thank you for your precious suggestions. We reply to your questions, modifying the manuscript text in red. Kindly regards.

Manuscript Title: Green Recycling for Polypropylene Components by Material Extrusion

  1. It would be better if the title included more details about the research.

Response: The authors prefer to maintain this title.

  1. It is best to state the most important results of the manuscript quantitatively at the end of the introduction.

Response: The results were qualitatively reported in the introduction.

  1. The research innovation is not highlighted at the end of the introduction.

Response: The innovation was reported in the introduction.

  1. It is better to display Figure 1 hierarchically.

Response: The figure items were hierarchically positioned.  

  1. The selection criteria for printing parameters such as nozzle diameter and layer height are unknown.

Response: The printing parameters were in the text, highlighted in red. 

  1. It would be better to add the semi-crystalline percentage to Table 1.

Response: The crystallization degree was added in Table 1.

  1. The selection of parameters in Table 3 requires discussion and reference.

Response: The discussion was added to the text. The reference was present in the text. 

  1. It would be better to merge Figures 9 and 10.

Response: The two figures were merged.

  1. The values in Table 4 require deviation from the standard.

Response: The table was updated.

  1. In Figure 9, it can be seen that the samples are not properly attached to the substrate and the separation between the layers is observed. If possible, add a left view and a front view of the samples to Figure 9.

Response: The figure was modified to show the separation from the heating plate.

  1. It is better to 3D print large-sized samples to better characterize the effects of shrinkage and interlayer separation.

Response: The MEX large specimens are not currently evaluated. This observation is a good point for further research.

Reviewer 2 Report

Comments and Suggestions for Authors

The conclusions obtained in the work are very interesting from the point of view of the use of recycled materials for additive manufacturing, specifically MEX technology.

Considering this objective, it would have been interesting to compare more than one recycled PP. And since it is stated in the conclusions (lines 419-421, pg.13) that "Adding PE to PP not only improved the quality of the printed parts..., but it also provided the framework for economically MEX-based manufacturing of recycled PP" it would be necessary to verify that this actually happens with other types of recycled PP. And above all, a planned study of controlled mixtures of PE and PP should be carried out to effectively evaluate the influence of PE on these improvements with respect to virgin PP. Therefore, what is indicated for further research about to optimize the PP/PE blend is considered interesting (line 424).

In line 75 (pg.2) it is indicated that MEX thermoplastic waste recycling shows a decrease in melt flow rate after one or multiple rounds of recycling. Please confirm that it is not the opposite, as MFR tends to increase (as molecular weight decrease) when reprocessing the material.

In line 219 it is suggested to replace the nomenclature FFF with MEX, according to the Terminology AM standard referenced in [6].

Finally, it is considered that 3 repetitions for the tensile tests are insufficient (see line 390, page 12), or at least, authors should include the standard deviation of the results in Table 4 (line 404, page 12), to assess whether more repetitions are necessary (at least 5) or if the average provided are representative values of the results.

Author Response

The conclusions obtained in the work are very interesting from the point of view of the use of recycled materials for additive manufacturing, specifically MEX technology.

  1. Considering this objective, it would have been interesting to compare more than one recycled PP. And since it is stated in the conclusions (lines 419-421, pg.13) that "Adding PE to PP not only improved the quality of the printed parts..., but it also provided the framework for economically MEX-based manufacturing of recycled PP" it would be necessary to verify that this actually happens with other types of recycled PP. And above all, a planned study of controlled mixtures of PE and PP should be carried out to effectively evaluate the influence of PE on these improvements with respect to virgin PP. Therefore, what is indicated for further research about to optimize the PP/PE blend is considered interesting (line 424).

Response: The evaluation of different PP/PE grades is ongoing. The results are currently under evaluation but not ready for the present manuscript.

  1. In line 75 (pg.2) it is indicated that MEX thermoplastic waste recycling shows a decrease in melt flow rate after one or multiple rounds of recycling. Please confirm that it is not the opposite, as MFR tends to increase (as molecular weight decrease) when reprocessing the material.

Response: The sentence was modified.

  1. In line 219, it is suggested that the nomenclature FFF be replaced with MEX, according to the Terminology AM standard referenced in [6].

Response: The nomenclature was corrected.

  1. Finally, it is considered that 3 repetitions for the tensile tests are insufficient (see line 390, page 12), or at least, authors should include the standard deviation of the results in Table 4 (line 404, page 12), to assess whether more repetitions are necessary (at least 5) or if the average provided are representative values of the results.

Response: The standard deviations were added to the table. 

Reviewer 3 Report

Comments and Suggestions for Authors

Dear authors,  

the work is devoted to an important topic: recycling and reusing plastic waste, which is especially relevant for 3D printing technology. Polypropylene is one of the most widely used plastic materials.

The manuscript is quite well organized, clear and easy to read.

However, some questions and comments have arisen regarding the work. Please find the details below.

1.       Line 106. It is the reviewer's opinion that Figure 1 may not be necessary. The reviewer believes that its inclusion to manuscript is redundant.

2.       Line 115. I have a suspicion that there is a typo in the name of the used DSС device. The manufacturer's website only has information about 404 F1 Pegasus model, not 403. Please check.

3.       Lines 127, 160 etc. The work contains some typos related to dimensions (units of measurement).

4.       Table 1. Are the experimentally obtained and presented in the table values typical for PP? It seems to me that it would be reasonable to compare them with the values known from the literature. See, for example, sources

https://analyzing-testing.netzsch.com/en/polymers-netzsch-com/commodity-thermoplastics/pp-isotactic-polypropylene

https://onlinelibrary.wiley.com/doi/full/10.1002/adem.201900796

https://iopscience.iop.org/article/10.1088/1757-899X/412/1/012070/pdf

or any others at your discretion

In the opinion of the reviewer, there are some discrepancies between your values from Table 1 (Tmelting, Melting Enthalpy) and what I can find in the literature.

5.       The captions to figures 3 and 4 should be rewritten. The data presented are FT-IR spectra from which plastic materials, what is their source?

6.       Page 7, line 254 There is an error in the numbering of figures. This is should fifth figure, and as a result, the numbering in the manuscript will need to be adjusted accordingly.

7.       My main remark. Table 4. Any mechanical testing is always a statistical question. How many samples were tested? What are the experimental errors of the measured mechanical characteristics? To what extent are the obtained values reproducible and how do they compare with the data known from the literature? In the opinion of the reviewer, the Young's modulus obtained in the work is quite small, the typical values are from 1300 to 1800 MPa,  why?

8.       Figure 8. What is the dimension of the horizontal axis (abscissa axis)?

Sincerely,

Author Response

The manuscript is quite well organized, clear and easy to read.

However, some questions and comments have arisen regarding the work. Please find the details below.

  1. Line 106. It is the reviewer's opinion that Figure 1 may not be necessary. The reviewer believes that its inclusion to manuscript is redundant.

Response: The authors prefer to keep the figure. Another reviewer asked to keep it and modify the orientation.

  1. Line 115. I have a suspicion that there is a typo in the name of the used DSС device. The manufacturer's website only has information about 404 F1 Pegasus model, not 403. Please check.

Response: The DSC model was corrected in 404 F1 Pegasus.

  1. Lines 127, 160 etc. The work contains some typos related to dimensions (units of measurement).

Response: The typos were corrected.

  1. Table 1. Are the experimentally obtained and presented in the table values typical for PP? It seems to me that it would be reasonable to compare them with the values known from the literature. See, for example sources
  1. https://analyzing-testing.netzsch.com/en/polymers-netzsch-com/commodity-thermoplastics/pp-isotactic-polypropylene
  2. https://onlinelibrary.wiley.com/doi/full/10.1002/adem.201900796
  3. https://iopscience.iop.org/article/10.1088/1757-899X/412/1/012070/pdf
  4. or any others at your discretion…

Response: The reference used was “PP polypropylene” - https://doi.org/10.1016/B978-1-895198-47-8.50144-2. The results were coherent with the values reported in this chapter. This reference was added to the manuscript.

  1. In the opinion of the reviewer, there are some discrepancies between your values from Table 1 (Tmelting, Melting Enthalpy) and what I can find in the literature.

Response: There were some errors in reporting melt enthalpy values in Table 1 and Figure 2. The results were corrected.

  1. The captions to figures 3 and 4 should be rewritten. The data presented are FT-IR spectra from which plastic materials, what is their source?

Response: The figure captions were updated.

  1. Page 7, line 254 There is an error in the numbering of figures. This is the fifth figure, and as a result, the numbering in the manuscript will need to be adjusted accordingly.

Response: The figure numbering was updated.

  1. My main remark. Table 4. Any mechanical testing is always a statistical question. How many samples were tested? What are the experimental errors of the measured mechanical characteristics? To what extent are the obtained values reproducible and how do they compare with the data known from the literature? In the opinion of the reviewer, the Young's modulus obtained in the work is quite small, the typical values are from 1300 to 1800 MPa,  why?

Response: The typical values of Young's modulus reported by the Netzsch document refer to injection-molding parts. The MEX parts had lower values due to a process that introduces porosity and interplay strength due to raster deposition. 

  1. Figure 8. What is the dimension of the horizontal axis (abscissa axis)?

Response: The horizontal axis length of Figure 9 (old Figure 8 after renumbering) is 20 mm, as reported in the scale in the same figure. 

Reviewer 4 Report

Comments and Suggestions for Authors

The subject is worthy of investigation and the conclusions are supported by the results.  Title is concise, accurate and informative. Abstract is a concise statement of the aims of the research, the work carried out and the conclusions. Introduction, Materials and method, Results and discussions are properly addressed. The characterization of the materials was very well described. The work is easy to read and understand. The graphic interpretations are intuitive and very well done. The paper has a real scientifc character. The subject is interesting with real applicability for industry.

Author Response

Dear Reviewer,

thank you for your precious suggestions. 

Kindly regards

Round 2

Reviewer 1 Report

Comments and Suggestions for Authors

It is best to state the most important results of the manuscript quantitatively at the end of the introduction. (In value or percent form)

The selection of parameters in Table 3 requires discussion and reference. (Why layer height is 0.2 for example)

Author Response

1. It is best to state the most important results of the manuscript quantitatively at the end of the introduction. (In value or percent form).

Response: The quantitative results were added to the manuscript (in red).

2. The selection of parameters in Table 3 requires discussion and reference. (Why layer height is 0.2 for example)

Response: Table 3 shows the values for filament making. Discussion and references were added.

The layer height refers to the MEX process. The 0.2 mm value was chosen to balance print quality and speed. Lower values improved quality but enormously increased print time. The opposite occurred with an increase in layer height.

Round 3

Reviewer 1 Report

Comments and Suggestions for Authors

NA